# Impact of the Coronavirus Disease 2019 Pandemic on the Management and Outcomes of ST-Segment Elevation Myocardial Infarction Patients: A Retrospective Cohort Study

**DOI:** 10.3390/medicina61030422

**Published:** 2025-02-27

**Authors:** Cheng-Han Yang, Yu-Jen Lin, Shi-Ying Gao, Wei-Chen Chen, Chung-Hsien Chaou

**Affiliations:** 1Department of Emergency Medicine, Chang Gung Memorial Hospital, Linkou Branch, Taoyuan 333, Taiwan; hanhan@cgmh.org.tw (C.-H.Y.); qoocue039@cgmh.org.tw (Y.-J.L.); s78092359@cgmh.org.tw (S.-Y.G.); jan323ice@cgmh.org.tw (W.-C.C.); 2College of Medicine, Chang Gung University, Taoyuan 333, Taiwan; 3Chang Gung Medical Education Research Center, Taoyuan 333, Taiwan

**Keywords:** severe acute respiratory syndrome coronavirus 2, coronavirus disease 2019, ST-segment elevation myocardial infarction, acute coronary syndrome, time-sensitive conditions

## Abstract

*Background and Objectives:* The coronavirus disease 2019 pandemic presented unprecedented challenges in balancing infection control measures with the timely management of ST-segment elevation myocardial infarction (STEMI), a time-sensitive condition. This study investigates the pandemic’s effects on STEMI management times and outcomes at a high-volume medical center in Taiwan. *Materials and Methods:* A retrospective analysis of 1309 STEMI patients was conducted at Chang Gung Memorial Hospital between 2017 and 2022. Patients were divided into pre-pandemic and pandemic groups. Measurement outcomes include in-hospital mortality rate, management times (e.g., door-to-balloon time), the rates of intra-aortic balloon pump (IABP) and/or veno-arterial extracorporeal membrane oxygenation (VA-ECMO) usage, mechanical ventilation, inotropic support, and the length of intensive care unit (ICU) and hospital stay. Kaplan–Meier survival analysis and statistical comparisons were performed to assess temporal trends and prognostic outcomes. *Results:* No significant difference in in-hospital mortality was observed between pre-pandemic (5.85%) and pandemic (7.03%) groups (*p* = 0.45). The pandemic group experienced longer management times, including door-to-cath arrival (*p* = 0.0335) and door-to-balloon time (*p* = 0.014), although all times remained below the 90 min threshold. Quality improvements during the first outbreak allowed the institution to handle higher case volumes during subsequent waves without further delays. Ninety-day survival analysis showed no significant disparity between groups (*p* = 0.3655). *Conclusions:* Pandemic-related delays in STEMI management were effectively mitigated through workflow optimization, preventing significant increases in mortality rates. This study highlights the adaptability of healthcare systems in responding to crises while maintaining quality care for time-sensitive emergencies. Future multicenter studies could provide broader insights into global STEMI management strategies under pandemic conditions.

## 1. Introduction

The coronavirus disease 2019 (COVID-19) outbreak, caused by severe acute respiratory syndrome CoV 2 (SARS-CoV-2), began in December 2019 in Wuhan, China, and rapidly escalated into a global health crisis. Declared a pandemic by the World Health Organization on 11 March 2020, it disrupted healthcare systems worldwide, with over 250 million cases and 5 million deaths reported globally by November 2021. In Taiwan, 15,253 deaths and 8.8 million cases were documented between 2019 and 2022, with early containment measures effectively reducing transmission and maintaining low case numbers [1,2,3,4]. Globally, the pandemic overwhelmed healthcare systems, diverting resources to COVID-19 care while delaying or deferring non-COVID-19 services. These disruptions exposed systemic vulnerabilities, particularly in managing critical conditions like ST-segment elevation myocardial infarction (STEMI) [5]. SARS-CoV-2 infection significantly increased the risk of arterial and venous thrombotic complications, doubling myocardial infarction (MI) risk within the first seven days of diagnosis [6]. Patients with STEMI and concurrent COVID-19 experienced worse outcomes, including markedly higher rates of major adverse cardiovascular events within 6 months [7].

During the COVID-19 pandemic, many patients in North America appeared hesitant to seek hospital care. Reports from the United States, Europe, China, Japan, South India, and South Africa indicated significantly lower hospitalization rates for acute MI, including both STEMI and non-STEMI [8,9,10,11,12,13]. Factors contributing to this decline included reduced referral rates, patient reluctance due to fear of contracting COVID-19 in healthcare settings, insufficient insurance coverage, and financial barriers to treatment and hospitalization [8,9,11,12,13]. Delays in seeking medical care were notable, with the symptom onset-to-door time in Taiwan increasing by 27% in 2020 compared to 2019 [14].

The pandemic posed unique challenges in STEMI care, including balancing the risks of staff exposure during primary percutaneous coronary intervention (PPCI) for COVID-19-positive patients against the increased morbidity and mortality associated with conservative management or fibrinolytic therapy [15]. Despite these obstacles, regions with high medical accessibility ensured that patients with suspected STEMI were promptly evaluated, with PPCI remaining the predominant treatment strategy. Registry data from Taiwan indicated a reduction in the median door-to-device time from 96 min in 2010 to 66 min in 2020, even during the pandemic [16].

Delayed reperfusion in STEMI patients is linked to worse outcomes, including larger MIs, heart failure, arrhythmias, and increased mortality [17]. While reduced door-to-balloon times have improved outcomes [18,19], patient-related delays—from symptom onset to first medical contact—remain a major challenge. Total ischemic time, the interval from symptom onset to revascularization, is a critical determinant of outcomes [20,21,22]. During the pandemic, delays in STEMI care led to increased morbidity and mortality, with prolonged ischemia worsening outcomes despite stable door-to-balloon times [14,23,24]. Notably, the existing literature has primarily focused on isolated time measurements, with limited correlation to follow-up prognostic outcomes. This study is the first in Taiwan to analyze data from the COVID-19 outbreak periods. Our study highlights the correlation between prognosis and time variations before and after an outbreak. This study examines the impact of the COVID-19 pandemic on STEMI care in a modern urban medical system, aiming to provide insights for improving management during future global health crises.

## 2. Materials and Methods

### 2.1. Study Design

This retrospective cohort study was conducted in Chang Gung Memorial Hospital, Linkou Branch (LCGMH) between October 2022 and January 2025. The study site is the largest single medical center in Taiwan, with a 3600-bed capacity and an annual emergency department (ED) visit of 160,000 patients. Patients’ data were anonymized before being included in the study. This study was conducted in accordance with the STROBE guidelines and was approved by the Hospital Ethics Committee on Human Research of Taiwan’s Chang Gung Medical Foundation (Institutional Review Board No. 202201575B0) and was qualified for a waiver of informed consent.

### 2.2. Patient Sampling

All adult patients who visited LCGMH between August 2017 and July 2022 and met the following inclusion criteria were included in the study: (1) presentation with chest pain-related symptoms at triage, (2) an initial electrocardiogram (ECG) suggestive of STEMI, and (3) a final diagnosis of STEMI (International Classification of Diseases 10th Revision code: I21.3) at discharge. Patients were excluded if the ECG interpretation by a consulting cardiologist was not consistent with STEMI, if the patient or their family declined cardiac catheterization, if the patient was discharged or transferred against medical advice (AMAD) prior to cardiac catheterization, or if cardiac catheterization findings did not indicate coronary artery disease. Patients were categorized into two groups: Group 1 represented the period before the COVID-19 pandemic (August 2017 to December 2019), and Group 2 represented the period after the onset of the COVID-19 pandemic (January 2020 to July 2022). A flowchart of the inclusion and exclusion process is presented in Figure 1.

### 2.3. Data Collection and Outcome Measurement

This study utilized data from the largest multi-institutional database in Taiwan, the Chang Gung Research Database (CGRD). The CGRD is a de-identified database derived from the regularly and systemically collected electronic medical records. Variables included are demographic information (e.g., age and sex), vital signs recorded during ED triage, and past medical history. Additional information gathered included relevant laboratory results, specific event time points and management timelines (e.g., door-to-ECG time and door-to-balloon time), and prognostic indicators such as catheterization findings, in-hospital mortality, critical interventions (e.g., mechanical ventilation [MV], intra-aortic balloon pump [IABP], and veno-arterial extracorporeal membrane oxygenation [VA-ECMO]), length of hospital stay, and length of intensive care unit (ICU) stay. The main measurement outcome of the study was the in-hospital mortality rate. The CGRD is linked to the Taiwan National Health Insurance Research Database. The patient’s cause of death, which was used for reimbursement, can be linked to CGRD individual data. Secondary measurement outcomes included the rates of IABP usage, VA-ECMO usage, MV, and inotropic support, as well as the length of hospital stay and duration of ICU admission. Data on the cumulative confirmed cases of COVID-19, obtained from the Taiwan Centers for Disease Control (CDC), were also analyzed.

### 2.4. Statistical Analysis

Continuous variables were presented as mean (standard deviation [SD]) or median (interquartile range [IQR; Q1–Q3]), while categorical variables were expressed as count (percentage). Differences between the two groups were analyzed using Student’s *t*-test or the Wilcoxon signed-rank test for continuous variables and the Chi-square test or Fisher’s exact test for categorical variables, where appropriate. This study also examined the relationship between management timelines and the COVID-19 pandemic, specifically assessing whether the pandemic period influenced management timelines and patient outcomes. To evaluate survival rates between the two groups, the Kaplan–Meier method was used to calculate survival rates and generate survival curves, and the log-rank test was employed to compare the survival curves. Statistical significance was defined as *p* < 0.05. All statistical analyses were performed using SAS version 9.4 (SAS Institute, Cary, NC, USA).

## 3. Results

This study included a total of 1309 patients who visited LCGMH between August 2017 and July 2022. The patients were divided into Group 1 (before the pandemic), comprising 598 patients, and Group 2 (during the pandemic), comprising 711 patients. The basic characteristics were comparable between the groups (Table 1). The primary outcome, in-hospital mortality, showed no statistical difference (Group 1: 5.85% vs. Group 2: 7.03%, *p* = 0.45). In terms of the secondary outcomes, while the VA-ECMO rate was significantly higher in Group 2 (0.5% vs. 2.25%, *p* = 0.0163), the other outcomes including the rate of IABP (*p* = 0.23), rate of MV (*p* = 0.72), rate of inotropic support (*p* = 0.43), length of hospital stay (*p* = 0.13), and ICU days (*p* = 0.11) were similar between the groups. For the analysis of management times, Group 2 had a significantly longer triage completion time (*p* = 0.028), cardiologist consultation time (*p* < 0.0001), arriving-cath-lab time (*p* = 0.0335), and door-to-balloon time (*p* = 0.014) (Figure 2). However, door-to-ECG time showed no significant differences (*p* = 0.068).

Figure 3 illustrates the trend of the monthly median door-to-ECG, door-to-cath, and door-to-balloon times during pre-COVID-19 and COVID-19 pandemic periods, contrasted by the log national COVID-19 case count. It can be observed that, in 2020, an initial drop in management times was noted at the beginning of the COVID-19 outbreak in China. While the outbreak did not yet involve Taiwan, ED visits decreased, and as a result, the median management time decreased. Afterward, the first outbreak period in early 2021 indeed led to significant delays in several management times. However, the last outbreak in 2022, although there were significantly more cases, did not further increase the management times. The detailed management times are presented in Figure 2.

A Kaplan–Meyer time-to-event curve of the 90-day mortality, stratified by pre-COVID-19 (Group 1, 2017–2019) and COVID-19 pandemic (Group 2, 2020–2022) periods, is presented in Figure 4. The event was defined as cardiovascular event-related deaths. There was no significant difference between the pre-pandemic and pandemic periods curves by log-rank test (*p* = 0.3655).

## 4. Discussion

This study analyzed treatment time intervals for STEMI patients across different stages of the COVID-19 pandemic. Linkou Chang Gung Memorial Hospital, with a total capacity of 3715 beds, stands as the largest medical institution in Taiwan, also in Southeast Asia [25]. The gender distribution in both groups, with male proportions exceeding 80%, aligns with the typical prevalence of STEMI in Taiwan. Previous epidemiological studies have shown that STEMI predominantly affects males in this region, which supports the representativeness of our study population. This consistency enhances the applicability of our findings to the broader Taiwanese STEMI patient cohort [26]. Significant differences were observed between the pre-pandemic and pandemic periods in door-to-complete triage, door-to-cardiologist consultation, door-to-ready for cath, door-to-cath arrival, and door-to-balloon times. Despite these delays, none of the average times exceeded the 90 min guideline, which likely contributed to the lack of a statistically significant difference in in-hospital mortality rates [27]. However, door-to-cath time was significantly prolonged during the pandemic, primarily due to infection control measures. Rapid COVID-19 testing added time for sample collection, transport, processing, and reporting. For COVID-19-positive patients, specialized cath rooms required preparation, disposable equipment coverings, and thorough post-procedure disinfection. Quality management revealed that delays in rapid testing and sample transportation were major contributors. To address this, the hospital mandated immediate sample collection for patients with chest pain and suspected STEMI on ECG, with dedicated staff ensuring sample delivery to the lab within 3 min.

STEMI is widely recognized as a time-sensitive condition [27]. However, during the COVID-19 pandemic, infection control measures, government policies, and public fear of contracting the virus led to delayed medical care globally [28,29,30,31,32,33,34]. Government policies have played a critical role in infection control [3,35]. In Taiwan, nationwide measures included stringent visitor restrictions in hospitals, strict border controls, regular official media updates and press conferences, and the implementation of a name-based mask rationing system. Additionally, the country adopted Travel, Occupation, Contact, and Cluster (TOCC)-based rapid triage, outdoor clinics, and protective sampling devices. TOCC-based screening has been actively promoted in Taiwan for years, serving as a key strategy in infection control. Studies from various countries reported increased hospitalizations during peak pandemic periods, along with a significant rise in in-hospital mortality among STEMI patients [36,37,38,39,40,41,42]. Research from the United States specifically linked this increase in mortality to delays in seeking medical care [31]. Conversely, some other studies from different regions found no significant differences in in-hospital mortality rates [12,32,43,44,45,46]. Our findings highlight that while treatment delays during a pandemic are inevitable due to infection control measures, maintaining management times within guideline-recommended thresholds remains a crucial goal. The strategies implemented in our study, such as workflow optimization and early adaptation to evolving infection control protocols, provide a reference for balancing infection control with timely emergency care. However, the applicability of these strategies depends on local healthcare capacity, pandemic severity, and policy frameworks. Future research should explore how different regions can tailor these approaches to optimize time-sensitive emergency management during global health crises.

This study is the first in Taiwan to analyze data from the COVID-19 outbreak periods. Two outbreak periods were identified based on CDC case counts and Ministry of Health alert levels: May–July 2021 and April–July 2022 [47]. As shown in Figure 3 and Table 1, treatment times during the first outbreak significantly increased compared to pre-pandemic levels due to mandatory pre-procedural COVID-19 testing. However, despite these delays, door-to-balloon time remained within the guideline-recommended window, ensuring appropriate STEMI care. Through workflow optimization, the second outbreak—despite a much higher surge in COVID-19 cases—did not lead to further treatment delays, demonstrating the hospital’s ability to sustain acute care services under pandemic pressures.

The COVID-19 pandemic had far-reaching impacts; however, this study focuses on the prognosis of the time-sensitive condition STEMI. Reviews of other studies highlight how varying national contexts, emergency medical systems, and hospital catheterization capacities contribute to differing outcomes [12,32,36,37,38,39,40,41,42,43,44,45,46]. Situated in a densely populated area, the study site serves as a major medical center in the region, and its findings may offer valuable insights for other nations. When time-sensitive emergencies intersect with infection control protocols, it is crucial to perform emergency treatments within the critical time window while implementing measures to prevent the transmission of infection to healthcare workers and other patients. Both are integral to effective patient care. This study reinforces confidence in managing STEMI patients during future pandemics. It confirms that dedicating a reasonable portion of the management time to infection control measures—while staying within the guideline-recommended door-to-balloon time—does not statistically increase patient mortality.

Finally, several limitations of this study must be acknowledged. First, as a single-center observational study, its sample size—though the largest in the country—may not match the scope of multicenter studies. Second, the study site was not significantly affected by strict lockdowns, which limits the study’s comparability to nations that experienced more stringent pandemic measures. Third, differences in patient characteristics, environments, and backgrounds between the pre-pandemic and pandemic periods introduce confounding factors that cannot be fully controlled. Lastly, as a retrospective study, patient selection was not randomized, which may introduce selection bias and affect the interpretation of the results.

## 5. Conclusions

The COVID-19 pandemic significantly impacted the management times for STEMI patients, particularly leading to an increase in door-to-balloon time due to mandatory infection control measures. However, despite these delays, treatment times remained within standard therapeutic guidelines, and no significant increase in in-hospital mortality or decline in three-month survival rates was observed. During the second outbreak, treatment times stabilized even as COVID-19 cases surged exponentially, reflecting the benefits of process optimization from the first outbreak. While infection control measures pose inherent challenges to timely STEMI management, proactive workflow adjustments and enhanced familiarity with protocols can help mitigate their impact, strengthening system resilience in future health crises.

## Figures and Tables

**Figure 1 medicina-61-00422-f001:**
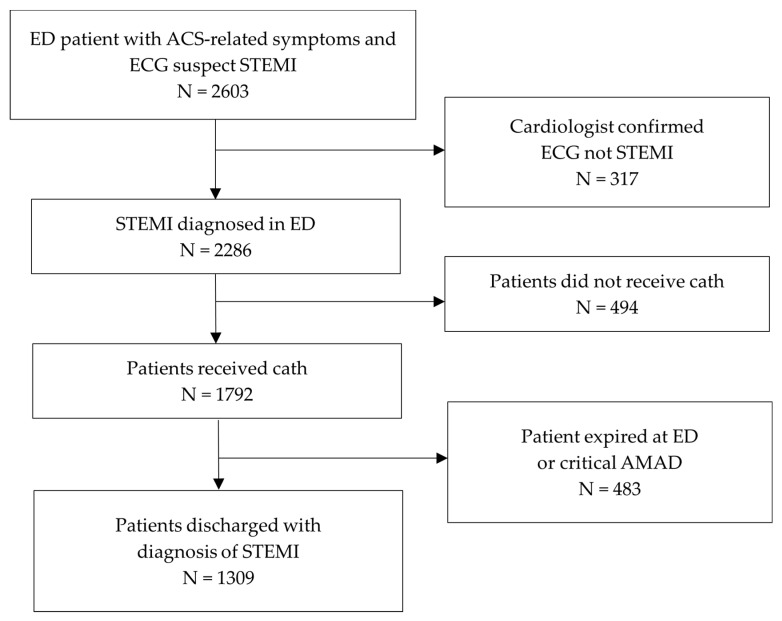
Inclusion and exclusion flowchart. ED—emergency department; ACS—acute coronary syndrome; ECG—electrocardiogram; STEMI—ST-segment elevation myocardial infarction; AMAD—against medical advice.

**Figure 2 medicina-61-00422-f002:**
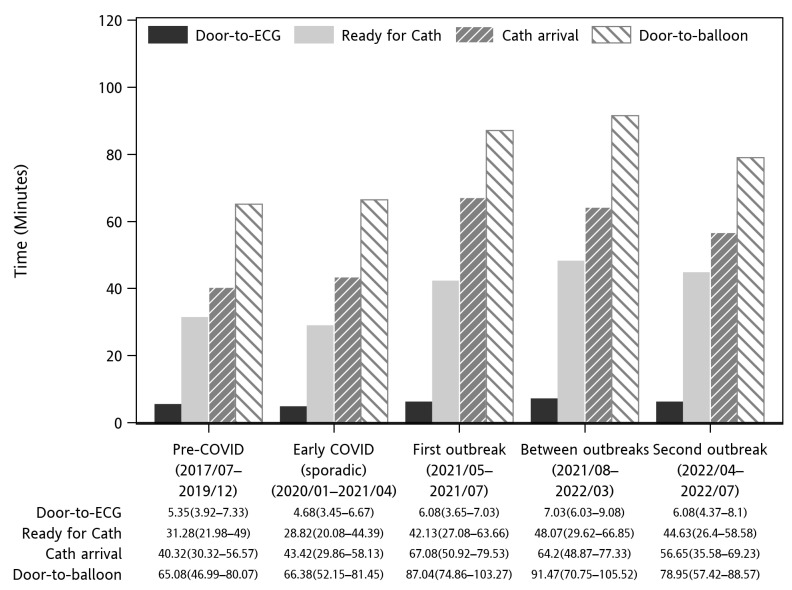
Different management times (minutes) during different periods of the pandemic.

**Figure 3 medicina-61-00422-f003:**
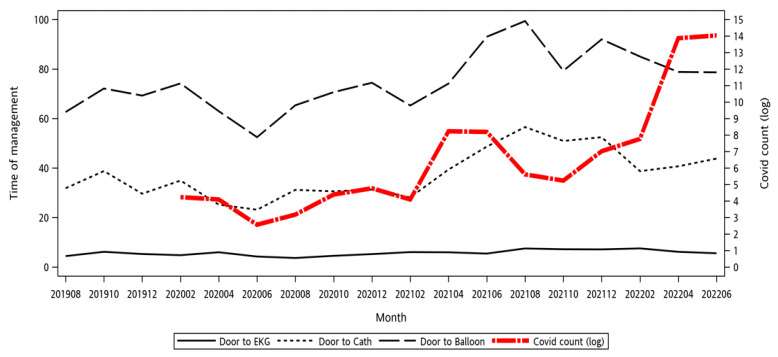
Monthly median management times and COVID-19 count (log) plot.

**Figure 4 medicina-61-00422-f004:**
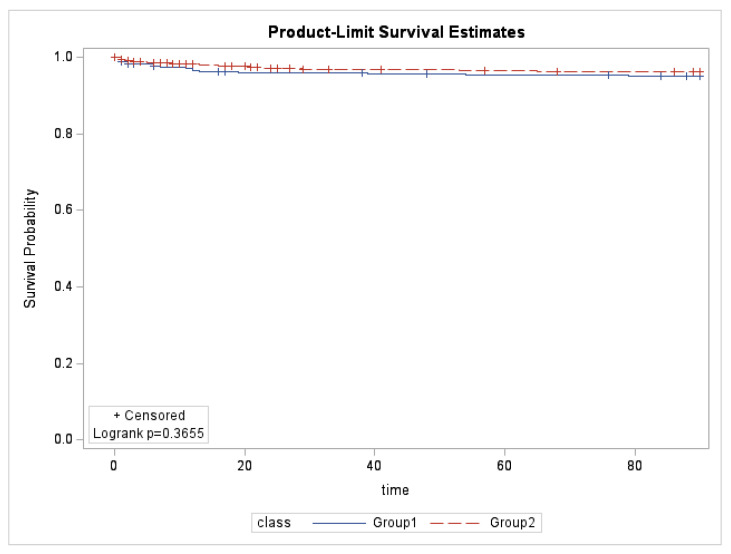
Kaplan–Meyer time-to-event curves of the 90-day mortality, stratified by pre-COVID-19 (Group 1, 2017–2019) and COVID-19 pandemic (Group 2, 2020–2022) periods. The event was defined as cardiovascular event-related deaths.

**Table 1 medicina-61-00422-t001:** Comparison of basic characteristics of emergency department ST-segment elevation myocardial infarction patients before and during the pandemic ^A^.

Variables	Total (N = 1309)	Group 1 2017–2019 (N = 598)	Group 2 2020–2022 (N = 711)	*p*-Value
Age ^¥^	62 (53–70)	60 (52–69)	63 (53–71)	0.0008 *
Male sex, n (%)	1068 (81.5)	483 (80.7)	585 (82.2)	0.5285
SpO_2_ (%) ^¶^	93.4 (16.4)	94.2 (14.4)	92.8 (18)	0.1299
Shock at triage, n (%)	172 (13.5)	93 (16.0)	79 (11.4)	0.0224 *
OHCA, n (%)	33 (2.5)	11 (1.8)	22 (3.1)	0.2041
Chest pain, n (%)	48 (3.6)	27 (4.5)	21 (2.9)	0.1771
Peak troponin (ng/mL) ^¶^	13.0 (23.8)	10.3 (21.5)	15.0 (25.2)	0.0047 *
CPK (U/L) ^¶^	1394.9 (1842.9)	1349.2 (1768.4)	1433.3 (1903.9)	0.4585
MB (ng/mL) ^¶^	8.1 (7.6)	9.0 (8.4)	7.4 (6.7)	0.0011 *
Past medical history				
Hypertension, n (%)	822 (62.8)	389 (65.0)	433 (60.9)	0.1362
Diabetes mellitus, n (%)	572 (43.7)	267 (44.6)	305 (42.9)	0.5616
Cigarette smoking, n (%)	597 (45.6)	282 (47.1)	315 (44.3)	0.3287
Prior PCI, n (%)	412 (31.4)	244 (40.8)	168 (23.6)	<0.0001 *
Prior CABG, n (%)	19 (1.4)	12 (2.0)	7 (0.9)	0.1908
Prior stroke, n (%)	178 (13.6)	74 (12.3)	104 (14.6)	0.2698
Outcomes				
In-hospital mortality, n (%)	85 (6.4)	35 (5.8)	50 (7.0)	0.4532
3-month mortality, n (%)	114 (8.7)	48 (8.0)	66 (9.2)	0.4812
IABP, n (%)	20 (1.5)	6 (1)	14 (1.9)	0.2330
VA-ECMO, n (%)	19 (1.4)	3 (0.5)	16 (2.2)	0.0163 *
MV, n (%)	43 (3.2)	18 (3.0)	25 (3.5)	0.7217
Inotropic support, n (%)	272 (20.7)	118 (19.7)	154 (21.6)	0.4309
Days of hospital stay ^¶^	8.1 (7.4)	7.8 (6.5)	8.4 (8.1)	0.1300
Days of ICU stay ^¶^	4.1 (4.2)	3.9 (3.5)	4.2 (4.8)	0.1107
Door-to-complete triage ^¥^	3.6 (2.1–6.5)	3.5 (1.9–6.1)	3.9 (2.2–6.7)	0.0284 *
Door-to-EKG ^¥^	5.6 (3.9–7.6)	5.3 (3.9–7.3)	5.9 (3.9–7.8)	0.0682
Door-to-consultation ^¥^	8.1 (5.4–11.4)	7.6 (4.9–10.5)	8.8 (5.9–11.9)	<0.0001 *
Door-to-ready for cath ^¥^	33.1 (22.1–53.4)	31.2 (21.9–49.0)	35.3 (22.4–56.8)	0.0163 *
Door-to-cath arrival ^¥^	51.6 (35.3–68.0)	40.3 (30.3–56.5)	53.2 (36.1–68.7)	0.0335 *
Door-to-balloon ^¥^	74.4 (56.6–89.4)	65.0 (46.9–80.0)	74.9 (57.4–89.7)	0.0143 *

^A^—All included patients underwent catheterization and were discharged with a diagnosis of STEMI. ^¶^—mean [SD], ^¥^—Median [IQR], and *—*p* value < 0.05. Abbreviations: SpO_2_—Saturation of Peripheral Oxygen; OHCA—Out-of-Hospital Cardiac Arrest; CPK—Creatine Phosphokinase; MB—Creatine Kinase-MB; PCI—Percutaneous Coronary Intervention; CABG—Coronary Artery Bypass Grafting; IABP—intra-aortic balloon pump; VA-ECMO—veno-arterial extracorporeal membrane oxygenation; Cath—Cathlab; ICU—intensive care unit; ECG—electrocardiogram.

## Data Availability

The data that support the findings of this study are available from the corresponding author upon reasonable request.

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
