# Peer review of "Impact of the Coronavirus Disease 2019 Pandemic on the Management and Outcomes of ST-Segment Elevation Myocardial Infarction Patients: A Retrospective Cohort Study"

_medicina, 2025, doi:10.3390/medicina61030422_

Round 1
Reviewer 1 Report
Comments and Suggestions for Authors
Knowledge void of this study in introduction is not clear. A single center has been considered
Comments on the Quality of English LanguageThe English could be improved to more clearly express the research.
Author Response
Dear editor and reviewer,
Thank you for taking the time to review our work. We are grateful for your valuable feedback and comments. We have made every effort to address the questions raised in detail, as requested, and have organized our responses point-by-point in the original order. In addition to the necessary changes based on the reviewers' suggestions, we also made some edits to streamline the content and improve readability. We hope these revisions have enhanced the quality of the article. The paragraphs in red are those that have been changed from the original article.
Major comments:
- Knowledge void of this study in introduction is not clear. A single center has been considered.
Response: Thank you for the reminder. The research gap exists in that current literature primarily focuses on isolated time measurements, with limited correlation to follow-up prognostic outcomes. Our study underscores the relationship between prognosis and time variations before and after an outbreak. We have added some descriptions to address the Knowledge void.
Introduction
…with limited correlation to follow-up prognostic outcomes. This study is the first in Taiwan to analyze data from the COVID-19 outbreak periods. Our study highlights the correlation between prognosis and time variations before and after an outbreak. This study examines…
- The English could be improved to more clearly express the research.
Response: Thank you for your comment. We have sent for professional English Editing and Proofreading. We hope that the quality of this manuscript has been improved.

Reviewer 2 Report
Comments and Suggestions for Authors
The manuscript titled “The Impact of Coronavirus Disease 2019…” presents a compelling study. However, there are a few aspects that need further clarification or refinement:
- As the authors highlighted, quality improvements during the initial outbreak allowed the institution to manage higher case volumes in subsequent waves without delays. However, it would be insightful to include data from the post-pandemic period (2022-2025) up until the sample collection time. This would help determine whether the same management flow has been consistently maintained after the pandemic.
- Both groups 1 and 2 had over 80% male patients. Is this in line with the typical prevalence of STEMI in the region? It would be helpful to clarify whether this gender distribution is representative of the local patient population.
- All abbreviations in the table legends (e.g., Table 1) should be expanded for clarity. This is important for ensuring that readers can fully understand the presented data without ambiguity.
- In the abstract, the authors mention three primary outcomes: management times (e.g., door-to-balloon time), in-hospital mortality, and 90-day cardiovascular mortality. However, in the results section, the second and third outcomes are presented as primary outcomes. I suggest these should be clarified and consistently presented as primary outcomes in Table 1, with a more thorough discussion of these outcomes before addressing secondary ones. Additionally, the door-to-balloon time is significantly different, with Group 2 showing a longer time than Group 1. This discrepancy needs to be discussed in more detail.
- The resolution of Figure 2 should be improved for better clarity. Higher resolution images are essential for precise data interpretation.
- In Table 2, the time presented is not specified in terms of units (days or months). It appears the information is somewhat redundant, so I recommend categorizing all primary and secondary outcome measures separately to streamline the presentation.
- The authors should consider discussing the measures taken by the government that helped maintain a non-significant difference between the two groups. This could be included in the second paragraph of the discussion section, providing context for the findings.
- The door-to-balloon time was significantly higher in Group 2. Despite this, the authors suggest that the hospital performed well during the pandemic. It would be beneficial to address how this increase in door-to-balloon time impacts the overall performance and whether other factors may have contributed to the perceived success.
- The conclusion should be updated to reflect the significant increase in door-to-balloon time during the pandemic. The authors should reframe their conclusion in light of this result and consider whether the hospital's performance during the pandemic aligns with their initial assertion of success.
Author Response
Dear editor and reviewer,
Thank you for taking the time to review our work. We are grateful for your valuable feedback and comments. We have made every effort to address the questions raised in detail, as requested, and have organized our responses point-by-point in the original order. In addition to the necessary changes based on the reviewers' suggestions, we also made some edits to streamline the content and improve readability. We hope these revisions have enhanced the quality of the article. The paragraphs in red are those that have been changed from the original article.
Major comments:
- As the authors highlighted, quality improvements during the initial outbreak allowed the institution to manage higher case volumes in subsequent waves without delays. However, it would be insightful to include data from the post-pandemic period (2022-2025) up until the sample collection time. This would help determine whether the same management flow has been consistently maintained after the pandemic.
Response: Thank you for your comment. The COVID-19 pandemic has led to significant changes in personnel, institutional policies, and computer system workflows in the post-COIVD era. Given these substantial differences, the current focus is on aspects that have remained relatively stable between the pre-pandemic and pandemic periods. This study serves as a preliminary step for a more comprehensive research plan in the future.
- Both groups 1 and 2 had over 80% male patients. Is this in line with the typical prevalence of STEMI in the region? It would be helpful to clarify whether this gender distribution is representative of the local patient population.
Response: Thank you for your comment. Yes, this result aligns with the current state of healthcare in Taiwan. We have added a paragraph to the Discussion section showing that our result is similar to the typical prevalence of STEMI from the Taiwan National Health Insurance Research Database.
Discussion:
…COVID-19 pandemic.In the current study, the gender distribution of both groups, with male proportions exceeding 80%, aligns with the typical prevalence of STEMI in Taiwan. Previous epidemiological studies have shown that STEMI predominantly affects males in this region, which supports the representativeness of our study population. This consistency enhances the applicability of our findings to the broader Taiwanese STEMI patient cohort. Significant differences were observed…
- All abbreviations in the table legends (e.g., Table 1) should be expanded for clarity. This is important for ensuring that readers can fully understand the presented data without ambiguity.
Response: Thank you for your reminder. We have made relevant changes as suggested.
- In the abstract, the authors mention three primary outcomes: management times (e.g., door-to-balloon time), in-hospital mortality, and 90-day cardiovascular mortality. However, in the results section, the second and third outcomes are presented as primary outcomes. I suggest these should be clarified and consistently presented as primary outcomes in Table 1, with a more thorough discussion of these outcomes before addressing secondary ones. Additionally, the door-to-balloon time is significantly different, with Group 2 showing a longer time than Group 1. This discrepancy needs to be discussed in more detail.
Response: Thank you for your comment. In our pre-specified study design, we identified in-hospital mortality as the main measurement outcome, while other variables, such as ICU stay duration, VA-ECMO usage, and mechanical ventilation rate, were considered supplementary measurement outcomes. To avoid misinterpretation, we have changed the wording in our abstract as suggested.
.
Abstract / Methods section
A retrospective analysis of 1,309 STEMI patients was conducted at Chang Gung Memorial Hospital from 2017 to 2022. Patients were divided into pre-pandemic and pandemic groups. Measurement outcomes include in-hospital mortality rate, management times (e.g., door-to-balloon time), the rates of IABP or VA-ECMO usage, mechanical ventilation, inotropic support, and the length of intensive care unit (ICU) and hospital stay. Kaplan-Meier survival analysis and statistical comparisons assessed temporal trends and prognostic outcomes.
Data collection and outcome measurement
The main measurement outcome of the study was the in-hospital mortality rate. The CGRD has a linkage to the Taiwan National Health Insurance Research Database. The patient’s cause of death, which was used for reimbursement, can be linked to CGRD individual data. Secondary measurement outcomes included the rates of IABP usage…
Dicussion, 1st paragraph:
in in-hospital mortality rates… Door-to-Cath time, however, was significantly prolonged during the pandemic, primarily due to infection control measures. Rapid COVID-19 testing added time for sample collection, transport, processing, and reporting. For COVID-positive patients, specialized Cath rooms required preparation, disposable equipment coverings, and thorough post-procedure disinfection. Quality management revealed that delays in rapid testing and sample transportation were major contributors. To address this, the hospital mandated immediate sample collection for patients with chest pain and suspected STEMI on ECG, with dedicated staff ensuring sample delivery to the lab within three minutes.
- The resolution of Figure 2 should be improved for better clarity. Higher resolution images are essential for precise data interpretation.
Response: Thank you for the reminder. We had improved the resolution of the figures.
- In Table 2, the time presented is not specified in terms of units (days or months). It appears the information is somewhat redundant, so I recommend categorizing all primary and secondary outcome measures separately to streamline the presentation.
Response: Thank you for the reminder. We have specified the time units. We have also made modifications to several tables and figures for a more direct visual comparison.
Table 2 changed to Figure 2:
Figure 2 Different time Consumption (Minutes) of management during different period of pandemic
- The authors should consider discussing the measures taken by the government that helped maintain a non-significant difference between the two groups. This could be included in the second paragraph of the discussion section, providing context for the findings.
Response: Thanks for the suggestion. Indeed, many governmental policy measures were implemented during the COVID period, significantly impacting the management of the outbreak. We have added relevant contents as suggested:
Discusison, 2nd paragraph:
……infection control. In Taiwan, nationwide measures included stringent visitor restrictions in hospitals, strict border controls, regular official media updates and press conferences, and the implementation of a name-based mask rationing system. Additionally, the country adopted Travel, Occupation, Contact, and Cluster (TOCC)-based rapid triage, outdoor clinics, and protective sampling devices. TOCC-based screening had been actively promoted in Taiwan for years, serving as a key strategy in infection control. Studies from various countries…
- The door-to-balloon time was significantly higher in Group 2. Despite this, the authors suggest that the hospital performed well during the pandemic. It would be beneficial to address how this increase in door-to-balloon time impacts the overall performance and whether other factors may have contributed to the perceived success.
Response: Thanks for your comment. The COVID pandemic had a significant impact on healthcare systems worldwide, especially in time-sensitive treatments. We believe that regardless of the outcomes, no one can truly claim to have "performed well," and we have avoided such descriptions from the manuscript.
In this result, we simply aim to highlight that, thanks to the efforts of medical teams and various response measures, STEMI patients were still able to receive treatment within the original time constraints. Additionally, although the door-to-balloon time was prolonged, there was no significant increase in mortality rates. We have further discussed this issue in the 3rd paragraph of the discussion section as follows:
Discussion, 3rd paragraph:
…As shown in Figure 3 and Table 1, treatment times during the first outbreak significantly increased compared to pre-pandemic levels due to mandatory pre-procedural COVID-19 testing. However, despite these delays, door-to-balloon time remained within the guideline-recommended window, ensuring appropriate STEMI care. Through workflow optimization, the second outbreak—despite a much higher surge in COVID-19 cases—did not lead to further treatment delays, demonstrating the hospital’s ability to sustain acute care services under pandemic pressures….
- The conclusion should be updated to reflect the significant increase in door-to-balloon time during the pandemic. The authors should reframe their conclusion in light of this result and consider whether the hospital's performance during the pandemic aligns with their initial assertion of success.
Response: Thanks for your comment. We have revised the conclusion to explicitly address the increase in management times and clarify its implications, ensuring that readers do not misinterpret our findings.
Conclusion
The COVID-19 pandemic significantly impacted the management times for STEMI patients, particularly leading to an increase in door-to-balloon time due to mandatory infection control measures. However, despite these delays, treatment times remained within standard therapeutic guidelines, and no significant increase in in-hospital mortality or decline in three-month survival rates was observed. During the second outbreak, treatment times stabilized even as COVID-19 cases surged exponentially, reflecting the benefits of process optimization from the first outbreak. While infection control measures pose inherent challenges to timely STEMI management, proactive workflow adjustments and enhanced familiarity with protocols can help mitigate their impact, strengthening system resilience in future health crises.

Reviewer 3 Report
Comments and Suggestions for Authors
peer review for the uploaded article:
Strengths
1. Relevance of the Topic:
- The article tackles an important and timely subject: the impact of the COVID-19 pandemic on the management and outcomes of STEMI patients, a critical issue in cardiovascular care.
- The study's focus on a time-sensitive condition and its implications for emergency and interventional cardiology is highly relevant to clinical practice and health system preparedness.
2. Robust Methodology:
- The use of a large dataset from a single high-volume center provides sufficient power to detect meaningful differences between pre-pandemic and pandemic groups.
- Clear inclusion and exclusion criteria, as well as adherence to STROBE guidelines, enhance the reliability and transparency of the study.
3. Comprehensive Data Analysis:
- The study employs appropriate statistical methods, including Kaplan-Meier survival analysis and comparisons of management times, to explore temporal and outcome trends.
- The detailed breakdown of management times (e.g., door-to-balloon, door-to-ECG) highlights critical delays and their potential implications.
4. Actionable Insights:
- The findings underscore the importance of workflow optimization in maintaining quality care for STEMI patients during health crises.
- The discussion of infection control measures and their integration into emergency protocols is highly practical and provides a roadmap for future pandemic preparedness.
Weaknesses
1. Limited Generalizability:
- As a single-center study in Taiwan, the findings may not be generalizable to regions with different healthcare infrastructures or stricter pandemic restrictions.
- The absence of data from multi-center studies limits the broader applicability of the conclusions.
2. Potential Selection Bias:
- The retrospective nature of the study introduces inherent biases, such as non-randomized patient selection, which may influence the interpretation of results.
3. Underexplored Variables:
- While the study focuses on key management times and mortality outcomes, it does not delve into other important factors, such as patient comorbidities or socioeconomic barriers, that could influence outcomes.
- Variations in provider-level factors, such as experience or adherence to guidelines during the pandemic, are not discussed.
4. Limited Contextual Comparison:
- The discussion does not sufficiently compare the findings to those from other countries or regions with similar or different COVID-19 responses, which could add valuable perspective.
Suggestions for Improvement
1. Expand the Discussion:
- Broaden the discussion to include comparisons with other international studies on STEMI care during the pandemic, highlighting similarities or differences in findings.
- Address how the results could inform global healthcare policies in managing time-sensitive emergencies during pandemics.
2. Enhance the Methodology:
- Consider a multi-center approach or collaboration with other institutions to strengthen the generalizability of findings.
- Provide additional details on how patient selection biases were mitigated in the analysis.
3. Explore Additional Outcomes:
- Investigate other variables, such as patient-reported outcomes, long-term follow-up data, or the impact of socioeconomic factors on care access and outcomes.
- Analyze potential differences in outcomes based on COVID-19 status (positive vs. negative).
4. Improve Clarity in Results Presentation:
- While the results section is detailed, some tables and figures could be simplified or presented with more direct visual comparisons (e.g., bar graphs or trend lines) to enhance readability.
Decision
Major Revision
The article is well-structured, methodologically sound, and offers valuable insights into STEMI care during the COVID-19 pandemic. However, addressing the weaknesses outlined above and incorporating the suggested improvements would enhance its impact and generalizability.
Author Response
Dear editor and reviewer,
Thank you for taking the time to review our work. We are grateful for your valuable feedback and comments. We have made every effort to address the questions raised in detail, as requested, and have organized our responses point-by-point in the original order. In addition to the necessary changes based on the reviewers' suggestions, we also made some edits to streamline the content and improve readability. We hope these revisions have enhanced the quality of the article. The paragraphs in red are those that have been changed from the original article.
Reviewer C
Major comments:
- Expand the Discussion:
- Broaden the discussion to include comparisons with other international studies on STEMI care during the pandemic, highlighting similarities or differences in findings. Address how the results could inform global healthcare policies in managing time-sensitive emergencies during pandemics.
Response: Thank you for your comment. We will rewrite the discussion to broaden discussion with other international studies and how our result could inform global healthcare policies.
Discussion, 2nd paragraph:
…in-hospital mortality among STEMI patients. Research from the United States specifically linked this increase in mortality to delays in seeking medical care. Conversely, some other studies from different regions found no significant differences in in-hospital mortality rates. Our findings highlight that while treatment delays during a pandemic are inevitable due to infection control measures, maintaining management times within guideline-recommended thresholds remains a crucial goal. The strategies implemented in our study,...
- Enhance the Methodology:
- Consider a multi-center approach or collaboration with other institutions to strengthen the generalizability of findings.
Response: Thank you for your suggestion. Given the current limitations in electronic medical records, future research initiatives will consider a multi-center collaboration to enhance generalizability. However, our patient cohort closely aligns with nationwide demographic characteristics reported by the Taiwan Centers for Disease Control(CDC), and as the largest medical institution in Taiwan, also in Southeast Asia, our single-center study retains a high degree of representativeness.
- Provide additional details on how patient selection biases were mitigated in the analysis.
Response: Thank you for your comment. To mitigate patient selection bias, the study used strict inclusion/exclusion criteria, ensuring all STEMI cases met standardized diagnostic and treatment protocols. Patient demographics were compared with national CDC data, confirming representativeness. Data were sourced from the Chang Gung Research Database, the largest electronic medical record database in Taiwan with multiple quality control measurements. Temporal stratification ensured consistency, and standardized management protocols minimized practice variation. Statistical adjustments accounted for confounders, ensuring reliable outcome comparisons.
- Explore Additional Outcomes:
- Investigate other variables, such as patient-reported outcomes, long-term follow-up data, or the impact of socioeconomic factors on care access and outcomes.
Response: Thank you for your wonderful suggestion. Indeed, some of the patient-reported outcomes and socioeconomic factors may be influential to the STEMI patient management. Unfortunately, many of the suggested variables are not available in this study. Plans for future prospective research based on the current study are underway.
- Analyze potential differences in outcomes based on COVID-19 status (positive vs. negative).
Response: Thank you for your comment. The number of STEMI cases with concurrent COVID-19 infection in this study was limited (N=5 vs. COVID-negative STEMI cases: N=706). Given the small sample size, the comparison lacks generalizability and, therefore was not presented after our discussion.
- Improve Clarity in Results Presentation:
- While the results section is detailed, some tables and figures could be simplified or presented with more direct visual comparisons (e.g., bar graphs or trend lines) to enhance readability.
Response: Thank you for the suggestion. We have improved the resolution of the figures and also made modifications to several tables and figures for a more direct visual comparison.
Table 2 changed to Figure 2:

Round 2
Reviewer 3 Report
Comments and Suggestions for Authors
no further comments